# Clinical Presentation, Microbiological Characteristics, and Their Implications for Perioperative Outcomes in Xanthogranulomatous Pyelonephritis: Perspectives from a Real-World Multicenter Practice

**DOI:** 10.3390/pathogens12050695

**Published:** 2023-05-10

**Authors:** Vineet Gauhar, José Iván Robles-Torres, Marcelo Langer Wroclawski, Hegel Trujillo-Santamaría, Jeremy Yuen Chun Teoh, Yiloren Tanidir, Abhay Mahajan, Nariman Gadzhiev, Deepak Ragoori, Santosh Kumar, Arvind Ganpule, Pankaj Nandkishore Maheshwari, Luis Roberto García-Chairez, Joana Valeria Enrriquez-Ávila, Juan Francisco Monzón-Falconi, Antonio Esqueda-Mendoza, Juan Pablo Flores-Tapia, Hugo Octaviano Duarte-Santos, Mudasir Farooq, Venkat Arjunrao Gite, Mriganka Mani Sinha, Bhaskar K. Somani, Daniele Castellani

**Affiliations:** 1Department of Minimally Invasive Urology, Ng Teng Fong General Hospital, Singapore 609606, Singapore; vineetgaauhaar@gmail.com; 2Urology Department, Hospital Universitario “Dr. José Eleuterio González,” Monterrey 64460, Mexico; ivan.robles25@live.com (J.I.R.-T.); roberto_garcia7@hotmail.com (L.R.G.-C.); jova.enrriquez@gmail.com (J.V.E.-Á.); 3Department of Urology, Hospital Israelita Albert Einstein, BP—A Beneficência Portuguesa de São Paulo, Sao Paulo 05562-900, SP, Brazil; urologia.marcelo@gmail.com; 4Faculdade de Medicina do ABC, Santo André 09060-870, SP, Brazil; 5Department of Urology, Hospital Covadonga, Córdoba 94560, Mexico; htsantamaria@gmail.com; 6S.H. Ho Urology Centre, Department of Surgery, Prince of Wales Hospital, The Chinese University of Hong Kong, Hong Kong 999077, China; jeremyteoh@surgery.cuhk.edu.hk; 7Department of Urology, School of Medicine, Marmara University, Istanbul 34854, Turkey; yiloren@yahoo.com; 8Department of Urology, Mahatma Gandhi Mission’s Medical College and Hospital, Aurangabad 431003, India; drabhaymahajan@gmail.com; 9Urology Department, Saint Petersburg State University Hospital, 199034 Saint Petersburg, Russia; nariman.gadjiev@gmail.com; 10Department Urology, Asian Institute of Nephrology and Urology, Hyderabad 500082, India; drragoori@gmail.com; 11Department Urology, Christian Medical College, Vellore 632004, India; drsksingh@hotmail.com (S.K.); mudasirfarooq@gmail.com (M.F.); 12Department of Urology, Muljibhai Patel Urological Hospital, Ahemadabad 387001, India; doctorarvind1@gmail.com; 13Department of Urology, Fortis Hospital Mulund, Mumbai 400078, India; dr.maheshwaripn@gmail.com; 14Department of Urology, Hospital Regional de Alta Especialidad de la Penisula de Yucatán, Mérdida 97133, Mexico; jfrancisco.falconi@hotmail.com (J.F.M.-F.); antonioesqueda@hotmail.com (A.E.-M.); uromid@gmail.com (J.P.F.-T.); 15Department of Urology, Hospital Municipal Dr. Moysés Deutsch (M’Boi Mirim), Sao Paulo 04849-030, SP, Brazil; hugosantos90@gmail.com; 16Department Urology, Grant Medical College & Sir JJ hospital, Mumbai 400008, India; balajigite@yahoo.com; 17Department of Urology, University Hospital Southampton NHS Foundation Trust, Southampton SO16 6YD, UK; mrigankamani@gmail.com (M.M.S.); bhaskarsomani@yahoo.com (B.K.S.); 18Urology Unit, Azienda Ospedaliero-Universitaria Delle Marche, Università Politecnica delle Marche, 60126 Ancona, Italy

**Keywords:** xanthogranulomatous pyelonephritis, urine culture, microbiology, urinary pathogens, prognosis

## Abstract

Xanthogranulomatous pyelonephritis (XGP) is an uncommon chronic granulomatous infection of renal parenchyma. XGP is often associated with long-term urinary tract obstruction due to stones and infection. We aimed to analyze the clinical, laboratory, and microbial culture profiles from bladder and kidney urine of patients who were diagnosed with XGP. Databases of patients with histopathological diagnosis of XGP from 10 centers across 5 countries were retrospectively reviewed between 2018 and 2022. Patients with incomplete medical records were excluded. A total of 365 patients were included. There were 228 (62.5%) women. The mean age was 45 ± 14.4 years. The most common comorbidity was chronic kidney disease (71%). Multiple stones were present in 34.5% of cases. Bladder urine culture results were positive in 53.2% of cases. Kidney urine culture was positive in 81.9% of patients. Sepsis and septic shock were present in 13.4% and 6.6% of patients, respectively. Three deaths were reported. *Escherichia coli* was the most common isolated pathogen in both urine (28.4%) and kidney cultures (42.4%), followed by *Proteus mirabilis* in bladder urine cultures (6.3%) and *Klebsiella pneumoniae* (7.6%) in kidney cultures. Extended-spectrum beta-lactamases producing bacteria were reported in 6% of the bladder urine cultures. On multivariable analysis, urosepsis, recurrent urinary tract infections, increased creatinine, and disease extension to perirenal and pararenal space were independent factors associated with positive bladder urine cultures. On multivariable analysis, only the presence of anemia was significantly more frequent in patients with positive kidney cultures. Our results can help urologists counsel XGP patients undergoing nephrectomy.

## 1. Introduction

It is estimated that the number of microbes in and on the human body is 10 times the total number of human cells, and collectively these influence our physiology during health and illness. Humans are seen as “superorganisms” where human and microorganisms live symbiotically [1]. Understanding the relationship between microbes and disease alongside with its impact on health and health systems has wide-ranging implications for human beings [2]. The impact of microorganisms as pathogens in renal infections is important from a causative, preventive, and management perspective. Xanthogranulomatous pyelonephritis (XGP) is uncommon chronic granulomatous damage of renal parenchyma, often associated with long-term urinary tract obstruction due to stones and recurrent infections. The association between upper urinary tract stones and XGP has been reported in 69% of patients, with 48% of them having staghorn calculi [3].

XGP peak of incidence is at age 50–70 years, with a 2:1 female:male ratio. XGP has an incidence of 1.4 cases per 100,000 people per annum [3].

Urine cultures most often reveal *Escherichia coli* and *Proteus mirabilis*, but sometimes uncommon and opportunistic pathogens too [4].

Computed tomography is the mainstay of diagnostic imaging. Both antibiotics and surgery are the cornerstones of treatment [5], and in a recent review by Harley et al. [3] 38 of the 40 included studies enrolled only patients with a histological diagnosis of XGP after undergoing radical or partial nephrectomy, showing that surgery was the standard of treatment. However, conservative management with antibiotics alone [6] can achieve good results.

The primary aim of our study was analyzing the clinical, laboratory, and microbial culture profiles from bladder and kidney urine of 365 patients who were diagnosed to have XGP in different parts of the world. The secondary aim was performing a subanalysis to see if the presence of positive urine cultures from the bladder and kidney had any bearing on or association with clinical and surgical outcomes, including mortality.

## 2. Materials and Methods

Databases of patients from 10 centers across 5 countries were retrospectively reviewed for patients with XGP confirmed by histopathological diagnosis after nephrectomy between 2018 and 2022. Patients not undergoing nephrectomy, without urine or kidney cultures, or with incomplete medical records were excluded. Ethics committee approval was obtained at each center recruited, and anonymized pooled data were then analyzed under the scrutiny of the Asian Institute of Nephrology and Urology Ethics Board Committee (AINU 14/2022). All patients signed an informed consent form to gather their anonymized data.

The following information was gathered upon hospital admission: age, gender, comorbidities, clinical characteristics, and laboratory workup at presentation (complete blood count, blood chemistry, and bladder urine culture). We considered the following cutoff values: anemia <12 g/dL hemoglobin, leukocytosis >12,000/μL white blood cells, thrombocytopenia <150,000/μL platelet, increased creatinine as serum creatinine level ≥1.2 mg/dL, and hyperglycemia as serum glucose level >200 mg/dL [7]. Data on urine and kidney cultures were acquired for analysis based on the guidelines of the Clinical and Laboratory Standards Institute. Identification of isolates was obtained by matrix-assisted laser desorption ionization time-of-flight mass spectrometry (MALDI-TOF MS) [8]. The production of ESBL was performed with a double-disk sensitivity test.

The quick Sepsis-Related Organ Failure Assessment (qSOFA) score was used to assess the risk of in-hospital mortality. [9]. A bladder urine culture was done on urine collected per urethra by patient voiding or from a bladder catheter.

For kidney urine culture, a sample was obtained from a preexisting or newly inserted nephrostomy tube on admission. Urinary drainage for initial relief of ureteral obstruction was obtained by a percutaneous nephrostomy or double J stent. The choice of drainage was entirely up to the surgeon’s preference and available resources at the place of practice.

Nephrectomy timing and approaches were also evaluated. Early nephrectomy was defined as ≤48 h upon admission, whereas delayed nephrectomy was considered if >48 h had passed. Approaches were classified as transperitoneal open surgery, retroperitoneal open surgery, or laparoscopy.

Data on the extension of XGP was recorded as described by the Malek clinical–radiological classification based on CT findings: (i) type 1: disease limited to the kidney; (ii) type 2: disease extension to perinephric space; and (iii) type 3: disease extension to paranephric space [10].

### Statistical Analysis

Categorical variables are described as frequencies and percentages. Continuous variables are described using medians and interquartile ranges. Univariable analysis was performed based on clinical, sociodemographic, biochemical, microbiological, and radiological variables to determine the association with positive urine and kidney cultures. Significant variables on univariable analysis were further analyzed in a multivariable model to identify independent factors related to the isolation of organisms in urine and kidney cultures. Multivariable analysis was performed with logistic regression to find the independent variables associated with the presence of positive cultures. Data are presented as odds ratio (OR) and 95% confidence interval (CI). Statistical tests were performed using SPSS for Windows, version 20.0 (IBM Corp., Armonk, NY, USA). Statistical significance was set at *p* < 0.05.

## 3. Results

A total of 365 patients met the inclusion criteria for analysis. Table 1 shows the characteristics of the study population.

There were 228 (62.5%) women. The mean age was 45 ± 14.4 years. Chronic kidney disease (CKD) was the most common comorbidity (71%), followed by hypertension (29%) and diabetes mellitus (28.5%). Almost 23% of patients had a history of recurrent urinary tract infections. Renal colic was the most frequent presenting symptom (39.7% of patients). Fever was present in 142 (38.9%) patients, whilst sepsis and septic shock were present in 49 (13.4%) and 24 (6.6%) patients, respectively. Bladder urine culture was positive in 185 (53.2%) patients, whereas kidney urine culture was positive in 118 (81.9%) cases. Multiple stones were present in 34.5% of cases. Half the patients (52.6%) presented with a double J stent or percutaneous nephrostomy tube already inserted. Early nephrectomy was performed in 192 (52.6%) patients. Three deaths occurred.

Table 2 shows the microbiological profile of urine and kidney cultures.

*Escherichia coli* was the most common isolated pathogen in both urine (28.4%) and kidney cultures (42.4%), followed by *Proteus mirabilis* in bladder urine cultures (6.3%) and *Klebsiella pneumoniae* (7.6%) in kidney cultures. Extended-spectrum beta-lactamases (ESBL) producing bacteria were reported only in 6% of the bladder urine cultures. Microbiological profiles of other bacteria are presented in Table 2. There were 127 patients who had both urine and kidney cultures, and among these, 86 (67.72%) patients had matching cultures with the same profile (Table 3). *Escherichia coli* was the most common isolated pathogen (50.0%), followed by *Klebsiella pneumoniae* (8.1%). In 20 (23.3%) patients both cultures were sterile.

On univariable analysis, urosepsis, recurrent urinary tract infections, leukocytosis, elevated creatinine, disease extended to peri- and pararenal space. Clavien–Dindo ≥ 3, local injuries, and admission to intensive care units were significantly associated with positive bladder urine cultures (Table 3). On multivariable analysis, urosepsis (*p* < 0.001, OR 3.94 [1.747–8.905]), recurrent urinary tract infections (UTI) (*p* < 0.001, OR 4.79 [2.373–9.674]), increased creatinine (*p* = 0.008, OR 2.54 [1.271–5.069]), disease extension to perirenal (*p* = 0.006, OR 2.004 [1.223–3.284]), and pararenal space (*p* = 0.027, OR 2.036 [1.082–3.830]) were independent factors associated with positive bladder urine cultures (Table 4).

On multivariable analysis, only the presence of anemia was significantly associated with positive kidney cultures (*p* = 0.02, OR 3.428 [1.210–9.707]) (Table 5).

## 4. Discussion

XGP was first described in 1916 by Schlagenhaufer. Oberling coined its current name in 1939, identifying the characteristic yellowish coloration of the affected kidney due to abundant lipid deposits within the cytoplasm of macrophages (xanthoma cells) on histopathological examination [11]. XGP is a clinical, radiological, and surgical dogma and is often referred to as a pseudotumor by its myriad presentations, but it is often associated with staghorn stones.

The challenge in treating renal infections with stones involves emergent intervention by percutaneous nephrostomy or double J stent to relieve an often-obstructed system [12], followed by a protracted medical management of the ensuing sepsis, often needing a nephrectomy when medical salvage fails [13]. With a deeper understanding of the disease pathology and microbiological profile, a better management strategy can be adopted.

### 4.1. Clinical Presentation and Microbiological Correlation in XGP

In our series, the mean age was 45 years and 62.5% of patients were women. This was similar to previous reports [3]. Overall, 13.4% of our patients presented with urosepsis and 22.7% of them reported having recurrent UTIs. Recurrent UTIs in women may be multifactorial and must be evaluated for kidney stone disease (KSD) [14]. Indeed, KSD and infections are reciprocally causal [15]. XGP may also be associated with upper urinary tract stones in up to 69% of cases [3]. In our series, staghorn stones were seen in 27.1% of cases and 69.7% of patients had some degree of preexisting CKD.

This association of recurrent UTI, female gender, KSD, and XGP is well known to precipitate sepsis, intensive care admission, the need for earlier nephrectomy, and often ensuing renal failure [16]. It is dubious to preclude aggressive antibiotic therapy irrespective of a negative urine culture. The literature reports that one-third of patients may have sterile urine [17]. In our series, 46.8% and 18.1% of urine and kidney cultures were sterile, respectively. Even if sterile urine is obtained at presentation, in patients not responding to conservative measures, despite internal or external urinary diversion, urologists must proactively adopt an aggressive multidisciplinary approach involving an expert radiologist, nephrologist, infectious disease specialist, and intensivist, or else the necessity for nephrectomy and risk of mortality increases [18]. We had three deaths, and all had positive urine and kidney cultures. To our knowledge, our series is the first to report the importance of performing a kidney culture, and we note that the latter is a more reliable source for establishing the causing pathogen and its microbiological sensitivity.

A possible diagnosis of XGP or emphysematous pyelonephritis must be considered in immunocompromised patients, especially whenever clinical presentation is atypical, current antibiotic therapy is not effective, and imaging shows features of dubious interpretation, since these patients have a very high risk of morbidity and mortality [19,20]. 71% of our patients had a history of CKD and diabetes was present in 28.5% of them. These diseases are known risk factors for poor prognosis in emphysematous pyelonephritis and XGP alike, and often patients need an early nephrectomy [21]. Overall, 72.8% of our patients had anemia, and like other reports, this is the commonest laboratory finding [3,16]. This could be due to underlying CKD and sepsis [22].

A 36.9% transfusion rate in our study reiterates its importance in perioperative planning. On multivariable analysis of factors associated with positive kidney cultures, anemia has a significant association. Whether this is incidental or has a prognostic significance and how this information can be better utilized for the management of renal infections remains to be ascertained.

### 4.2. Radiological Findings and Microbiological Correlations

Computed tomography is the gold-standard imaging modality for XGP staging. Classic findings include a loss of normal renal parenchyma, an enlarged kidney, paradoxical contracted scarred renal pelvis, and dilated calyces resulting in a multiloculated appearance. This combination resembles the paw print of a bear (bear’s paw sign) and is often associated with the presence of staghorn stones [23]. In some cases, there is a perinephric extension with thickening of Gerota’s fascia with calcification [23]. The two common XGP presentations are diffuse (90%) and focal or tumefactive form (10%). The former is more common and staged by Malek and Elder into three different stages according to the extent of involvement in nearby tissue [10]. Harley et al. argued that this classification was rarely used in clinical practice and that no correlation can be made between stages and mortality rate [3]. Malek classification was available for all patients in our series, and there was a significant correlation with a positive urine culture. Perirenal (stage 2) and pararenal infection (stage 3) were more likely to occur or be present if both cultures are positive. Again, as stated earlier, this reiterates the need to have both bladder and kidney urine cultures, and we think that even if one is positive, appropriate antibiotic treatment can be tailored to patient prognosis and clinical condition. This information is vital for medical and surgical intervention, and we advocate its use as a prognostic tool, as possibly these patients need a more aggressive antibiotic regime pre- and postoperatively. We were unable to quantify what the antibiotic regime used in our series was, since this information was not available.

### 4.3. Operative Outcomes and Microbiological Correlation

Overall, 52.6% of patients in our series needed an early nephrectomy. Presence of a positive urine culture, indicating a higher probability of a septic patient, was directly and significantly associated with a higher probability of developing Clavien–Dindo grade 3 and above complications. Whilst not significant, a higher proportion of patients with positive kidney cultures had Clavien–Dindo grade 3 and above complications (30.5% vs. 23.1%) vis à vis those with a negative culture at surgery. This could be because our complications included all patients undergoing nephrectomy (early and delayed) and they might have already been on a higher antibiotic regimen. However, as surgical intervention itself is an independent predictive factor for complications and morbidity, it becomes imperative to look at all parameters that may aid in counseling patients before surgery [3,16].

### 4.4. Microbiological Profiles in Urine and Kidney Cultures in XGP

Understanding pathogenesis and its impact on a disease process is essential for developing proper management strategies. In an era of antibiotic resistance and superbugs [24], if we can start appropriate antibiotic treatment early, we could avoid the above-stated problems and save a patient from a life-threatening infectious disease. XGP is most commonly associated with *Proteus mirabilis* or *Escherichia coli* infection, and Pseudomonas species have also been implicated [3,25]. In our series, *Escherichia coli* was the most common isolated pathogen in both bladder and kidney urine cultures. This finding was in line with 16 of 20 included studies in Harley’s review, where Proteus species were most commonly cultured microorganisms in the remaining ones [3]. There is also a wide range of less common pathogens in the literature, including species Candida, Klebsiella, Enterobacter, Streptococcus, Corynebacterium, Morganella, and Pseudomonas aeruginosa [3]. We also isolated the abovementioned species, but interestingly, we also demonstrated that a polymicrobial infection was present in roughly 4% of our patients.

*Proteus mirabilis* and *Escherichia coli* are usually sensitive to various antibiotics, including first-generation cephalosporins and trimethoprim–sulfamethoxazole. Pseudomonas species have a narrower spectrum to which they are sensitive and may require the use of aminoglycosides, third-generation cephalosporins, or fluoroquinolones. In our study, the presence of *Pseudomonas aeruginosa* had a 4% incidence in bladder urine culture and 6.3% in kidney urine cultures. Whilst ours was a real-world practice where surgeons might have used different antibiotic regimens, we found that ESBL bacteria were present in 6.08% of bladder urine cultures. There were no reported ESBL producing bacteria in kidney urine cultures. Perhaps, coupled with the absence of an antibiotic regime, a limitation inherent to the retrospective nature of this study precludes further subset analysis to recommend the ideal antibiotic treatment to start at presentation, even before a culture is obtained. However, there was microbiological concordance in 76.7% of those patients who had both cultures with growing pathogens. This reiterates the importance of having a bladder urine culture in all XGP patients before starting antibiotics, since it commonly harbors the actual pathogen.

## 5. Conclusions

An insight into the clinical presentation from this large, global, multicenter real-world XGP study highlights for the first time the need to obtain both bladder and kidney urine cultures and importance of using Malek’s classification. Together with microbiological and clinical findings, our results can help urologists to counsel the prognostic expectations in patients undergoing nephrectomy.

## Figures and Tables

**Table 1 pathogens-12-00695-t001:** Demographic, clinical, radiological, and microbiological characteristics of the study population (n = 365).

**Demographics**	
Age; years mean ± SD	45 ± 14.4
Females; n (%)	228 (62.5)
Right side kidney; n (%)	184 (50.4)
Body mass index; kg/m^2^, mean ± SD	24.5 ± 5.2
Days of hospitalization; days, mean ± SD	9.2 ± 6.5
**Clinical presentation**	
Fever n (%) (>38.3 °C)	142 (38.9)
Renal colic; n (%)	145 (39.7)
Urosepsis; n (%)	49 (13.4)
Shock n (%) (medium arterial pressure < 60 mmHg)	24 (6.6)
**qSOFA score**	
0 points; n (%)	261 (71.5)
1 point; n (%)	54 (14.8)
2 points; n (%)	31 (8.5)
3 points; n (%)	11 (3)
**Comorbidities**	
Chronic kidney disease; n (%)	259 (71)
Hypertension; n (%)	106 (29)
Diabetes mellitus; n (%)	104 (28.5)
Recurrent urinary tract infections; n (%)	83 (22.7)
**Radiological characteristics**	
Pyonephrosis; n (%)	169 (46.3)
Ureteropelvic junction stone; n (%)	110 (30.1)
Multiple stones; n (%)	126 (34.5)
Staghorn stone; n (%)	99 (27.1)
Renal abscess; n (%)	90 (24.7)
**Laboratory workup**	
Anemia (hemoglobin < 12 g/dL)	265 (72.8)
Leukocytosis (leukocytes > 12,000/L)	120 (32.9)
Increased creatinine (creatinine > 1.2 mg/dL)	88 (24.1)
Hyperglycemia (glucose > 200 mg/dL)	48 (13.1)
Thrombocytopenia (platelets < 150,000/L)	25 (6.8)
**Malek classification**	
Limited to the kidney; n (%)	176 (48.2)
Extended to perirenal space; n (%)	124 (34)
Extended to pararenal space; n (%)	65 (17.8)
**Microbiological characteristics**	
Positive kidney cultures; n (%) ^1^	118 (81.9)
Urine and kidney cultures positive; n (%) ^2^	78 (61.4)
Positive bladder urine cultures; n (%) ^3^	185 (53.2)
**Previous urinary diversion**	
Percutaneous nephrostomy; n (%)	129 (35.3)
Double J stent; n (%)	63 (17.3)
Percutaneous drainage of abscess; n (%)	38 (10.4)
**Nephrectomy approach**	
Early nephrectomy; n (%)	192 (52.6)
Transperitoneal; n (%)	187 (51.2)
Retroperitoneal; n (%)	178 (48.8)
Laparoscopic; n (%)	32 (8.8)
**Perioperative findings**	
Operation time; minutes, mean ± SD	173.2 ± 65.5
Blood loss; mL mean ± SD	641.5 ± 791.9
Need of blood transfusions; n (%)	135 (36.9)
**Postoperative Complications (Clavien–Dindo)**	
Grade I; n (%)	51 (14)
Grade II; n (%)	46 (12.6)
Grade IIIa; n (%)	11 (3)
Grade IIIb; n (%)	19 (5.2)
Grade IVa; n (%)	38 (10.4)
Grade IVb; n (%)	25 (6.8)
Grade V; n (%)	3 (0.8)

^1^ Considering total kidney cultures (n = 144) as 100%; ^2^ considering total urine and kidney cultures (n = 127) as 100%; ^3^ considering total bladder urine cultures (n = 348) as 100%.

**Table 2 pathogens-12-00695-t002:** Microbiological profile of bladder urine and kidney cultures.

	Number (%)
**Bladder urine cultures (n = 348)**	
Sterile	163 (46.8)
*Escherichia coli*	99 (28.4)
*Proteus mirabilis*	22 (6.3)
*Klebsiella pneumoniae*	18 (5.2)
Polymicrobial	16 (4.6)
*Pseudomonas aeruginosa*	14 (4)
*Enterococcus faecalis*	6 (1.8)
Others ^1^	10 (2.9)
ESBL agents in urine; n (%)	21 (6.03)
**Kidney urine cultures (n = 144)**	
Sterile	26 (18.1)
*Escherichia coli*	61 (42.4)
*Klebsiella pneumoniae*	11 (7.6)
*Pseudomonas aeruginosa*	9 (6.3)
*Proteus mirabilis*	8 (5.6)
Polymicrobial	6 (4.2)
*Staphylococcus aureus*	6 (4.2)
*Candida* spp.	5 (3.4)
*Morgagnella morgagni*	5 (3.4)
Others ^2^	7 (4.8)

^1^* Candida* spp. (3), *Enterobacter cloacae* (2), *Staphylococcus aureus* (3), *Citrobacter freundii* (1), *Gardenela vaginalis* (1). ^2^
*Enterococcus faecalis* (2), *Enterobacter cloacae* (2), *Streptococcus constellatus* (1), *Corynebacterium* spp. (1), *Pasteurella pneumotropica* (1).

**Table 3 pathogens-12-00695-t003:** Microbiological profile of urine and kidney cultures in patients with matching results (n = 86).

	Number (%)
Sterile	20 (23.3)
*Escherichia coli*	43 (50.0)
*Klebsiella pneumoniae*	7 (8.1)
*Pseudomonas aeruginosa*	5 (5.8)
*Proteus mirabilis*	4 (4.6)
Polymicrobial	3 (3.6)
*Staphylococcus aureus*	2 (2.3)
*Candida* spp.	2 (2.3)

**Table 4 pathogens-12-00695-t004:** Univariable and multivariable analysis for clinical, biochemical, and radiological factors associated with positive bladder urine cultures in patients with xanthogranulomatous pyelonephritis (n = 348).

			Univariable Analysis		Multivariable Analysis	
	Positive Bladder Urine Culture (n = 185)	Negative Bladder Urine Culture (n = 163)	*p* Value	OR (CI 95%)	*p* Value	OR (CI 95%)
Demographics						
Age; years mean ± SD	46.6 ± 14.8	43 ± 15.7	0.08	-		
Females; n (%)	113 (61.1)	101 (62)	0.866	1.038 (0.673–1.600)		
Clinical Presentation						
Fever; n (%) (>38.3 °C)	64 (34.6)	68 (42.2)	0.206	0.755 (0.487–1.168)		
Urosepsis; n (%)	37 (20)	12 (7.4)	0.001	3.146 (1.579–6.268)	0.001	3.944 (1.747–8.905)
qSOFA ≥ 2 points; n (%)	26 (14)	14 (8.6)	0.072	1.870 (0.939–3.724)		
Comorbidities						
Diabetes mellitus; n (%)	56 (30.3)	45 (27.6)	0.585	1.138 (0.715–1.812)		
Hypertension; n (%)	64 (34.6)	40 (24.5)	0.051	1.626 (0.019–2.597)		
Chronic kidney disease; n (%)	128 (69.2)	118 (72.4)	0.512	0.856 (0.538–1.362)		
Recurrent urinary tract infections; n (%)	64 (34.6)	18 (11)	<0.001	4.261 (2.395–7.579)	<0.001	4.791 (2.373–9.674)
Radiological characteristics						
Renal abscess; n (%)	53 (28.6)	34 (20.9)	0.094	1.523 (0.929–2.497)		
Staghorn stone; n (%)	50 (27)	45 (27.6)	0.818	1.057 (0.657–1.700)		
Multiple stones; n (%)	68 (36.8)	53 (32.5)	0.407	1.206 (0.774–1.880)		
Laboratory workup						
Anemia (hemoglobin < 12 g/dL)	141 (76.2)	110 (67.5)	0.07	1.544 (0.964–2.473)		
Leukocytosis (leukocytes > 12,000/L)	75 (40.5)	41 (25.1)	<0.001	2.840 (1.762–4.579)	0.082	1.724 (0.934–3.183)
Thrombocytopenia (platelets < 150,000/L)	17 (9.2)	8 (4.9)	0.056	2.301 (0.961–5.507)		
Increased creatinine (creatinine > 1.2 mg/dL)	63 (34.1)	23 (14.1)	<0.001	3.143 (1.840–5.370)	0.008	2.539 (1.271–5.069)
Hyperglycemia (glucose > 200 mg/dL)	30 (16.2)	18 (11)	0.041	1.930 (1.022–3.645)	0.542	1.307 (0.552–3.093)
Malek classification						
Limited to the kidney; n (%)	70 (37.8)	94 (47.7)	Reference			
Extended to perirenal space; n (%)	73 (39.5)	46 (28.2)	0.002	2.131 (1.317–3.449)	0.006	2.004 (1.223–3.284)
Extended to pararenal space; n (%)	42 (22.7)	23 (14.1)	0.003	2.452 (1.352–4.447)	0.027	2.036 (1.082–3.830)
Clinical Outcomes and Complications						
Clavien–Dindo ≥ 3; n (%)	67 (36.2)	27 (16.6)	<0.001	2.860 (1.717–4.764)	0.51	1.360 (0.544–3.401)
Local injuries; n (%)	31 (16.8)	15 (9.2)	0.038	1.986 (1.030–3.829)	0.841	1.094 (0.456–2.626)
Vascular lesion	16 (8.6)	7 (4.3)	0.103	2.110 (0.846–5.265)		
Colonic lesion	10 (5.4)	6 (3.7)	0.443	1.495 (0.531–4.208)		
Pleural lesion *	7 (3.8)	1 (0.6)	0.072	6.371 (0.775–52.342)		
Duodenum lesion *	2 (1.1)	1 (0.6)	0.99	1.770 (1.159–19.708)		
Spleen lesion *	2 (1.1)	2 (1.2)	0.99	0.880 (1.123–6.317)		
Intensive care unit admission; n (%)	46 (24.9)	13 (8)	<0.001	3.818 (1.979–7.369)	0.051	2.634 (1.000–6.937)
Mortality; n (%)	3 (1.6)	0 (0)	0.063	0.525 (0.475–0.580)		

* Fisher’s exact test.

**Table 5 pathogens-12-00695-t005:** Univariable and multivariable analysis for clinical, biochemical, and radiological factors associated with positive kidney cultures in patients with xanthogranulomatous pyelonephritis (n = 144).

			Univariable Analysis		Multivariable Analysis	
	Positive Kidney Culture (n = 118)	Negative Kidney Culture (n = 26)	*p* Value	OR (CI 95%)	*p* Value	OR (CI 95%)
Demographics						
Age; years mean ± SD	46.01 ± 15.8	48.9 ± 15.4	0.395	-		
Females; n (%)	45 (38.1)	11 (42.3)	0.693	0.841 (0.355–1.991)		
Clinical Presentation						
Fever; n (%) (>38.3 °C)	38 (32.2)	8 (30.8)	0.78	1.140 (0.454–2.860)		
Urosepsis; n (%)*	25 (21.2)	2 (7.7)	0.111	3.226 (0.714–14.581)		
qSOFA ≥ 2 points; n (%) *	21 (17.8)	2 (7.7)	0.246	2.769 (0.607–12.643)		
Comorbidities						
Diabetes mellitus; n (%)	38 (32.2)	12 (46.2)	0.176	0.554 (0.234–1.313)		
Hypertension; n (%)	32 (27.1)	12 (46.2)	0.056	0.434 (0.182–1.038)		
Chronic kidney disease; n (%)	73 (61.9)	14 (53.8)	0.449	1.390 (0.591–3.272)		
Recurrent urinary tract infections; n (%)	32 (27.1)	5 (19.2)	0.405	1.563 (0.543–4.494)		
Radiological characteristics						
Renal abscess; n (%)	49 (41.5)	8 (30.8)	0.31	1.598 (0.643–3.969)		
Staghorn stone; n (%)	29 (24.6)	12 (46.2)	0.059	0.434 (0.180–1.047)		
Multiple stones; n (%)	43 (36.4)	7 (26.9)	0.356	1.556 (0.605–4.001)		
Laboratory workup						
Anemia (hemoglobin < 12 g/dL)	99 (83.9)	15 (57.7)	0.002	4.033 (1.598–10.182)	0.02	3.428 (1.210–9.707)
Leukocytosis (leukocytes > 12,000/L)	55 (46.6)	8 (30.8)	0.024	2.813 (1.118–7.074)	0.199	1.942 (0.705–5.357)
Thrombocytopenia (platelets < 150,000/L) *	13 (11)	1 (3.8)	0.186	4.276 (0.532–34.351)		
Increased creatinine (creatinine > 1.2 mg/dL)	32 (27.1)	9 (34.6)	0.475	0.720 (0.291–1.778)		
Hyperglycemia (glucose > 200 mg/dL)	15 (12.7)	4 (15.4)	0.99	1.109 (0.332–3.703)		
Malek classification						
Limited to the kidney; n (%)	38 (32.2)	15 (57.7)	Reference			
Extended to perirenal space; n (%)	44 (37.3)	6 (23.1)	0.045	2.895 (1.022–8.202)	0.136	2.436 (0.755–7.856)
Extended to pararenal space; n (%)	36 (30.5)	5 (19.2)	0.065	2.842 (0.937–8.624)	0.306	1.968 (0.538–7.199)
Clinical Outcomes and Complications						
Clavien–Dindo ≥ 3; n (%)	36 (30.5)	6 (23.1)	0.45	1.463 (0.542–3.950)		
Local injuries; n (%) *	19 (16.1)	4 (15.4)	0.99	1.056 (0.327–3.411)		
Vascular lesion *	9 (7.6)	1 (3.8)	0.69	2.064 (0.250–17.047)		
Colonic lesion *	7 (5.9)	2 (7.7)	0.665	0.757 (0.148–3.871)		
Pleural lesion *	4 (3.4)	0 (0)	0.99	0.814 (0.752–0.881)		
Duodenum lesion *	0 (0)	2 (7.7)	0.052	0.169 (0.117–0.243)		
Spleen lesion *	2 (1.7)	0	0.99	0.817 (0.756–0.883)		
Intensive care unit admission; n (%) *	20 (16.9)	2 (7.7)	0.367	2.449 (0.535–22.203)		
Mortality; n (%) *	3 (2.5)	0 (0)	0.99	0.814 (0.752–0.881)		

* Fisher’s exact test.

## Data Availability

Data will be provided upon reasonable request by the corresponding author.

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
