# Peer review of "Clinical Presentation, Microbiological Characteristics, and Their Implications for Perioperative Outcomes in Xanthogranulomatous Pyelonephritis: Perspectives from a Real-World Multicenter Practice"

_pathogens, 2023, doi:10.3390/pathogens12050695_

Round 1

Reviewer 1 Report

Congratulations on the present paper! Elaborating on this material reveals lots of effort in collecting and analyzing all the included data.

Xanthogranulomatous pyelonephritis is a rare disease that can hardly be managed by medical staff, which makes it a challenging and exciting topic.

Even though this paper is well documented, there are still some improvements to be made for future publishing.

1.     The Abstract needs massive revision. It is pretty embarrassing to see those spelling errors.

2.     The English language also needs revision for the entire document.

3.     In the  Materials and Methods section, you can include a table for the cut-off values of the blood tests.

4.     The qSOFA is an international score that needs no further explanation.

5.     In the Results, you simply repeat the data that can easily be observed in the table.

6.     Also, the data from the table can be classified in decreased order.

7. The Discussion section has a subsection named 4.2 Radiological findings. You didn’t make any personal data analysis or correlations in this section, and you only used the available data from the literature.

8.     You can also include in your study a subsection with less specific cases. You can use the following paper: 10.3390/pathogens11010081

9.     In the Conclusion section, you assume the main idea is to highlight the need to obtain both urine and kidney culture, and this idea should be extended more in the main text with proper correlations.

The manuscript needs major English revision. I suggest a native English speaker read it.

Author Response

We thank the reviewer for their constructive comments which helped us to improve our manuscript.

Reviewer #1

Congratulations on the present paper! Elaborating on this material reveals lots of effort in collecting and analyzing all the included data. Xanthogranulomatous pyelonephritis is a rare disease that can hardly be managed by medical staff, which makes it a challenging and exciting topic.

REPLY. We wish to thank for your nice comments on our study,

Even though this paper is well documented, there are still some improvements to be made for future publishing.

  1. The Abstract needs massive revision. It is pretty embarrassing to see those spelling errors.

REPLY: Thank you very much for your comment. We do apologies for that. Abstract has been revised

  1. The English language also needs revision for the entire document.

REPLY: Thank you very much for your comment. We do apologies for that. English has been extensively reviewed.

  1. In the Materials and Methods section, you can include a table for the cut-off values of the blood tests.

REPLY. Thank you very much for your comment. With the greatest respect of you comment, we believe that adding a Table with cut-off values in Methods section would add nothing more since cut-off values are already well presented.

  1. The qSOFA is an international score that needs no further explanation.

REPLY: Thank you very much for your comment. According to your valuable suggestion, we deleted the explanation of qSOFA score

  1. In the Results, you simply repeat the data that can easily be observed in the table.

REPLY. Thank you very much for your comment. Results section commonly reports the major findings of our study. Of course, this part duplicate data from Tables. This occurs in every scientific paper.

  1. Also, the data from the table can be classified in decreased order.

REPLY. Thank you very much for your comment. According to your valuable suggestion, we have ordered data in Table 1 in decreased order. We left Table 2 as it was because we believe that sterile urine should stay first and Others last despite their order.

  1. The Discussion section has a subsection named 4.2 Radiological findings. You didn’t make any personal data analysis or correlations in this section, and you only used the available data from the literature.

REPLY. Thank you very much for your comment. The main outcome of this paper was assessing the role of bladder and kidney cultures in XGP patients and not the role of radiological findings in XGP. This is why we did not discuss in depth radiological findings and, conversely, explored the association between XGP extension and positive cultures in two multivariable analysis. This point was implemented in section 4.2 as follows “Malek classification was available for all patients in our series and there was a significant correlation with a positive urine culture. Perirenal (stage 2) and pararenal infection (stage 3) were more likely to occur or be present if both cultures are positive. Again, as stated earlier, this reiterates the need to have both kidney and urine culture and we think that even if one is positive appropriate antibiotic treatment can be tailored to patients prog-nosis and clinical condition. This information is vital for medical and surgical intervention and we advocate its use as a prognostic tool as possibly these patients need a more ag-gressive antibiotics regime pre and post-operatively. We were unable to quantify what was the antibiotic regime used in our series since this information was not available.”

According to your suggestion we have implemented this part as follows

  1. You can also include in your study a subsection with less specific cases. You can use the following paper: 10.3390/pathogens11010081

REPLY: Thank you very much for suggesting to read this paper that we read with interested. We cited it in Introduction.

  1. In the Conclusion section, you assume the main idea is to highlight the need to obtain both urine and kidney culture, and this idea should be extended more in the main text with proper correlations.

REPLY. Thank you very much for your comment. This part has been implemented and now reads as follows “Understanding pathogenesis and its impact on a disease process is quintessential for developing proper management strategies. In an era of antibiotic resistance and superbugs [22], if we can start appropriate antibiotic treatment early, we could avoid the above-stated problems and save a patient from a life-threatening infectious disease. XGP is most commonly associated with Proteus mirabilis or Escherichia coli infection; Pseudomonas species have also been implicated [11,23]. A similar microbial profile was seen in our patients. Proteus mirabilis and Escherichia coli are usually sensitive to various antibiotics, including first-generation cephalosporins and trimethoprim-sulfamethoxazole. Pseudomonas species have a narrower spectrum to which they are sensitive and may require the use of aminoglycosides, third-generation cephalosporins, or fluoroquinolones. In our study presence of Pseudomonas aeruginosa was reported as a 4% incidence in urine culture and 6.3% in kidney cultures. Whilst ours was a real-world practice where surgeons might have used different antibiotic regimens, we found that ESBL bacteria were present in 6.08% of urine cultures. There were no reported ESBL producing bacteria in kidney cultures. Perhaps, coupled with the absence of an antibiotic regime, a limitation inherent to the retrospective nature of this study precludes further subset analysis to recommend the ideal antibiotic treatment to start at presentation even before a culture is obtained. However, there was a microbiological concordance in 76.7% of those patients who had both cultures with growing pathogens. This reiterates the importance of having a urine culture in all XGP patients before starting antibiotics since it commonly harbors the actual pathogen.”

Reviewer 2 Report

In the work entitled "Clinical presentation, microbiological characteristics and their impact on perioperative outcomes in Xanthogranuloma-tous Pyelonephritis: Perspectives from a real-world Multicenter practice", the authors analyze the data of patients diagnosed with XGP and try to find the relationship of urinary tract infections to clinical and surgical outcomes. The aim of the work is interesting and deals with an important problem, and the analysis was done well based on the inclusion of a group of members. Below are my notes and comments.

- the introduction is too general, the authors repeatedly mention in the text that XGP is uncommon, which immediately indicates that they should explain this problem to the readers in a more detailed way. More general information is contained in the discussion than in the introduction. What is the frequency of diagnosis? What is the current standard of treatment, is antibiotic therapy always indicated?

- In this age range, women have an increased risk of developing a urinary tract infection. Why were the patients not separated into two groups: men and women in the data presentation?

- what is known about concomitant urolithiasis. What is the origin of the stones, whether they were formed as a result of metabolic disorders or during infection. How important is this in XGP development?

- An important aspect discussed in the work is the microbiological profile. Too little comparative data on the presence and prevalence of specific microorganisms has been presented in the discussion. for example, there are records that mention P. mirabilis first

Author Response

In the work entitled "Clinical presentation, microbiological characteristics and their impact on perioperative outcomes in Xanthogranulomatous Pyelonephritis: Perspectives from a real-world Multicenter practice", the authors analyze the data of patients diagnosed with XGP and try to find the relationship of urinary tract infections to clinical and surgical outcomes. The aim of the work is interesting and deals with an important problem, and the analysis was done well based on the inclusion of a group of members. Below are my notes and comments.

- the introduction is too general, the authors repeatedly mention in the text that XGP is uncommon, which immediately indicates that they should explain this problem to the readers in a more detailed way. More general information is contained in the discussion than in the introduction. What is the frequency of diagnosis? What is the current standard of treatment, is antibiotic therapy always indicated?

REPLY. We would like to thank you for this comment. we have implemented introduction to your valuable comment. Introduction now reads as follows “It is estimated that the number of microbes in and on the human body is 10 times the total number of human cells and collectively, and these influence our physiology during health and illness. Humans are seen as “superorganisms” where human and microorganism live symbiotically. Understanding the relationship between microbes and disease alongside with its impact on health and health systems has wide-ranging implications for human beings. The impact of microorganisms as pathogens in renal infections is important from a causative, preventive, and management perspective. Xanthogranulomatous Pyelonephritis (XGP) is an uncommon chronic granulomatous damage of renal parenchyma, often associated with long-term urinary tract obstruction due to stones and recurrent infections. The association between upper urinary tract stones and XGP has been reported in 69% of patients, with 48% of them having staghorn calculi. XGP peak of incidence is at age 50–70 years, with a 2:1 female to male ratio. XGP has an incidence of 1.4 cases per 100.000 people per annum Urine cultures most often reveal Escherichia coli and Proteus mirabilis but sometimes uncommon and opportunistic pathogens too Computed tomography is the mainstay of diagnostic imaging. Both antibiotics and surgery are the cornerstones of its treatment [3] and in a recent review by Harley et al. 38 of the 40 included studies enrolled only patients with a histological diagnosis of XGP after undergoing radical or partial nephrectomy, showing that surgery was the standard of treatment. However, conservative management with antibiotics alone (reference: Korkes F, Favoretto RL, Broglio M, Silva CA, Perez MDC, Castro MG. Xanthogranulomatous pyelonephritis: clinical experience with 41 cases. Urology 2008; 71: 178–80) can achieve good results. The primary aim of our study was analyzing the clinical, laboratory, and microbial culture profiles from bladder and kidney urine of 365 patients who were diagnosed to have XGP in different parts of the world. The secondary aim was performing a sub analysis to see if the presence of positive urine cultures from the bladder and kidney had any bearing or association with clinical and surgical outcomes including mortality.”

- In this age range, women have an increased risk of developing a urinary tract infection. Why were the patients not separated into two groups: men and women in the data presentation?

REPLY: Thank you very much for your important comment. Commonly, data are presented from the whole series. The aim of our study was not comparing female versus male, this is another paper. That is why we presented in that way.

- what is known about concomitant urolithiasis. What is the origin of the stones, whether they were formed as a result of metabolic disorders or during infection. How important is this in XGP development?

REPLY. Thank you very much for your important comment. Unfortunately, we cannot provide data for metabolic disorders or infection origin of stones in our series because we did not gather. The importance of concomitant urolithiasis has been implemented in the introduction as follows “The association between upper urinary tract stones and XGP has been reported in 69% of patients, with 48% of them having staghorn calculi

- An important aspect discussed in the work is the microbiological profile. Too little comparative data on the presence and prevalence of specific microorganisms has been presented in the discussion. for example, there are records that mention P. mirabilis first

REPLY. Thank you very much for your important comment. In accordance with your valuable comment, we have implemented Discussion part (section 4.4) with the following paragraph “In our series, Escherichia coli was the most common isolated pathogen in both bladder and kidney urine cultures. This finding was in line with 16 out of 20 included studies in Harley’s review, where Proteus species were most commonly cultured microorganisms in the remaining ones. There is in literature also a wide range of less common pathogens including species Candida, Klebsiella, Enterobacter, Streptococcus, Corynebacterium, Morganella, and Pseudomonas aeruginosa. We also isolated the above-mentioned species but, interestingly, we also demonstrated that a polymicrobial infection was present in roughly 4% of our patients”.

Reviewer 3 Report

Summary: The authors analyzed the clinical, laboratory, and microbial culture profiles from bladder and kidney urine patients who were diagnosed to have XGP.

 Major comments:  A big number of patients were included for analysis, therefore, the results and findings are highly reliable. The reviewer would like to know the match rate between urine and kidney cultures.

Minor comments: In Abstract, delete "o" from" bladder and kidney urine o patients"

Author Response

We thank the reviewer for their constructive comments which helped us to improve our manuscript.

Reviewer #2

Summary: The authors analyzed the clinical, laboratory, and microbial culture profiles from bladder and kidney urine patients who were diagnosed to have XGP.

 Major comments:  A big number of patients were included for analysis, therefore, the results and findings are highly reliable. The reviewer would like to know the match rate between urine and kidney cultures.

REPLY. Thank you very much for your comment. According to your valuable suggestion we performed an analysis of concordance between the two cultures and reported results in a new Table (Table 3). We have also implemented Results section as follows “There were 127 patients who had both urine and kidney cultures and among these, 86 (67.72%) patients had matching cultures with same profile (Table 3). Escherichia coli was the most common isolated pathogen (50.0%), followed by Klebsiella pneumoniae (8.1%). In 20 (23.3%) patients both cultures were sterile.”

Minor comments: In Abstract, delete "o" from" bladder and kidney urine o patients"

REPLY. Thank you very much for your comment. That was a typo error. We do apologies for that. The sentence now reads as follows “We aimed to analyze the clinical, laboratory, and microbial culture profiles from bladder and kidney urine of patients who were diagnosed with XGP”

Round 2

Reviewer 1 Report

The recommended modifications were completed accordingly, so I suggested accepting the present paper.

English form was improved.

Reviewer 2 Report

I thank the authors for responding to comments and taking them into account in the preparation of the revised version of the article. I have no more comments.